# Left Ventricular Systolic Function Assessed by Speckle Tracking Echocardiography in Athletes with and without Left Ventricle Hypertrophy

**DOI:** 10.3390/jcm8050687

**Published:** 2019-05-15

**Authors:** Aleksandra Żebrowska, Rafał Mikołajczyk, Zbigniew Waśkiewicz, Zbigniew Gąsior, Katarzyna Mizia-Stec, Damian Kawecki, Thomas Rosemann, Pantelis T. Nikolaidis, Beat Knechtle

**Affiliations:** 1Department of Physiological and Medical Sciences, Academy of Physical Education, Mikołowska Street 72a, 40-065 Katowice, Poland; a.zebrowska@awf.katowice.pl (A.Ż.); r.mikolajczyk@awf.katowice.pl (R.M.); 2Department of Team Sports Games, Academy of Physical Education in Katowice, Mikołowska Street 72a, 40-065 Katowice, Poland; z.waskiewicz@awf.katowice.pl; 3Department of Sports Medicine and Medical Rehabilitation, Sechenov University, Moscow 119991, Russia; 4Department of Cardiology, School of Health Science, Medical University of Silesia, Katowice, Poland Ziołowa Street 47, 40-635 Katowice, Poland; zbgasior@gmail.com; 51st Department of Cardiology, School of Medicine Medical University of Silesia, Katowice, Poland Ziołowa Street 47, 40-635 Katowice, Poland; kmiziastec@gmail.com; 62nd Department of Cardiology, School of Medicine with the Division of Dentistry in Zabrze, Medical University of Silesia, Skłodowskiej, Curie 10 Street, 41-800 Zabrze, Poland; damian.kawecki@sum.edu.pl; 7Institute of Primary Care, University of Zurich, 8091 Zurich, Switzerland; thomas.rosemann@usz.ch; 8Exercise Physiology Laboratory, Thermopylon 7, 18450 Nikaia, Greece; pademil@hotmail.com; 9Medbase St. Gallen Am Vadianplatz, 9001 St. Gallen, Switzerland

**Keywords:** cardiac function, cycling, endurance

## Abstract

The aim of this study was to evaluate selected parameters of strain and rotation of the left ventricle (the basal rotation (BR) index, the basal circumferential strain (BCS) index, and the global longitudinal strain (GLS) of the left ventricle) in male athletes with physiological cardiac hypertrophy (LVH group), and athletes (non-LVH group) and non-athletes without hypertrophy (control group, CG). They were evaluated using transthoracic echocardiography and speckle tracking echocardiography before and after an incremental exercise test. The LVH group demonstrated lower BR at rest than the non-LVH group (*p* < 0.05) and the CG (*p* < 0.05). Physical effort had no effect on BR, nor was this effect different between groups (*p* > 0.05). There was a combined influence of LVH and physical effort on BR (F = 5.70; *p* < 0.05) and BCS (F = 4.97; *p* < 0.05), but no significant differences in BCS and GLS at rest between the groups. A higher BCS and lower GLS after exercise in the LVH group were demonstrated in comparison with the CG (*p* < 0.05). Left ventricular basal rotation as well as longitudinal and circumferential strains showed less of a difference between rest and after physical effort in subjects with significant myocardial hypertrophy. In conclusion, the obtained results may suggest that echocardiographic assessment of basal rotation and circumferential strain of the left ventricular can be important in predicting cardiac disorders caused by physical effort in individuals with physiological and pathological heart hypertrophy.

## 1. Introduction

Endurance training affects adaptive changes of the heart, causing hypertrophy, including an increase in the left ventricular mass and a proportional increase in the thickness of the posterior left ventricular wall and interventricular septum [1,2]. Adaptive changes include an improved contractility of the heart muscle, an increased coronary reserve, and beneficial changes to the vascular system of the skeletal muscles [3,4]. Several factors determine cardiac hypertrophy in competitive athletes, such as the type of muscle work [5], the intensity of training [6], the length of the training experience [7], and both sex and age [8]. For example, athletes subjected to the same training loads can have different discipline-specific levels of heart hypertrophy [7,9]. Differences in heart structure between individuals can also significantly influence the adaptation to high-intensity efforts; the correlation between a predisposition toward endurance efforts and genetically determined heart hypertrophy is well documented [10]. A few reports also suggested hormonal involvement, whose increased secretion during physical effort may affect cardiac muscle hypertrophy and left ventricle contractility in both animals [11] and humans [12].

It has recently been documented that the high hemodynamic load on the heart caused by physical training may alter echocardiogram-related parameters of heart deformation when they are determined by tracking acoustic markers (e.g., by speckle tracking echocardiography (STE)) [13,14]. When using STE to assess cardiac muscle deformation, the heart muscle is visualized by a reflected ultrasonic beam producing pixels as a result of interference, with a group of approximately 20–40 pixels creating a visible speckle [15,16]. STE enables assessment of the left ventricle as it undergoes deformation throughout the cardiac cycle across three planes of motion (i.e., longitudinal, radial, and circumferential). Furthermore, “twist mechanics” can also be determined [17,18,19].

Elite-level endurance athletes demonstrated differences in left ventricular strain and twist mechanics compared with untrained individuals [13], particularly in the magnitude of the radial strain (RS) and basal circumferential strain (BCS) of the left ventricle versus healthy subjects with a sedentary lifestyle [8,20]. Previous studies suggested that the differences in STE-derived parameters may be dependent on the sport, such that they can be categorized by low or high dynamic components [8,20]. The described myocardial biomechanics may have a potential role to characterize physiological hear hypertrophy (athlete′s heart) by itself or to distinguish it from hypertrophic cardiomyopathy [21].

Thus far, it is unclear whether physiological hypertrophy of the heart caused by endurance training could affect the systolic function of the left ventricle in competitive athletes. In addition, no previous work has assessed whether there is a correlation between the echocardiographic indicators of heart structure and selected variables of basal strain and rotation of the left ventricle in athletes with and without heart hypertrophy. Therefore, the aim of this study was to evaluate selected parameters of strain of the left ventricle in male athletes with physiological cardiac hypertrophy, as well as to compare them with athletes and non-athletes without hypertrophy.

## 2. Experimental Section

### 2.1. Subjects

Forty two of 52 male subjects who attended clinical examinations during the recruitment period met the inclusion criteria. Participants were assigned to the athletes group or the control group (CG): 36 patients were assigned to the athletes group, and 16 subjects were assigned to the CG. Four participants were excluded due to cardiac abnormalities (arrhythmia, prolapse of the mitral valve) and six for insufficient imaging quality, leaving 42 subjects for the final analyses (28 actively road cycling and 14 volunteers not practicing competitive sports). Athletes were recruited from elite cyclists’ sports associations in Poland. They were divided into groups based on a clinical finding of left ventricle hypertrophy (LVH) or not (non-LVH). The non-athletes volunteers with normal heart dimensions and a clinical finding of no LVH served as a control group (CG). The cardiovascular system and other clinical examinations of the participants were normal, and they did not have any systemic diseases. Subjects were excluded if they had a concurrent cardiovascular or respiratory disorder. Participants’ body mass and body composition were determined using bioelectrical impedance analysis (BIA; InBody Data Management System, Biospace, Korea). Age-matched healthy controls participated in recreational activities over four years, i.e., Nordic-walking; cycling; mountain tracking; or, in general, leading an active lifestyle. The trained subjects were characterized by a similar training status and a similar level of aerobic capacity (Table 1). All individuals were instructed to abstain from strenuous exercise 24 h prior to the study. No caffeine, antioxidants supplements, or alcohol were permitted during 48 h before the experiment. All participants gave their written informed consent to take part in the study. The study was approved by the local Bioethical Committee (University Ethics Committee decision N3/2010) and conducted in accordance with the Declaration of Helsinki of the World Medical Association.

### 2.2. Study Protocol

All subjects performed a standard incremental exercise test to measure their individual aerobic performance (Sport Excalibur, the Lode Company, Groningen, The Netherlands) in terms of maximum power (Pmax) and maximum oxygen uptake (VO_2_max). Prior to the exercise test, all subjects underwent baseline echocardiography. Resting echocardiographic examination was performed in all subjects in the morning after a night’s rest (Rest). STE-based echocardiographic examination was repeated in all subjects after an incremental stress test on a cycling ergometer (Post-Ex). The subjects underwent post-effort echocardiography after stabilization of the heart rate at about 150 bpm, i.e., in about the fifth minute of rest.

#### 2.2.1. Echocardiography

Echocardiography was performed with a GE Vivid E9 device (General Electric, Horten, Norway), which is equipped with a head frequency of 2.5 MHz. The resting echocardiogram was performed using the one-dimensional (M-mode), two-dimensional (2D), and Doppler methods. From the M-mode image obtained in the parasternal view under the control of the two-dimensional mode, primary measurements were made as follows: left ventricular end-diastolic diameter (LVEDd), left ventricular end-systolic diameter (LVESd), interventricular septum thickness (IVS), and posterior wall thickness (PWT) of the left ventricle. Mean values of three cardiac cycles were analyzed. We did not observe significant respiratory variation of flow velocities, and the test was performed during the expiration phase of respiratory cycle. On the basis of the obtained measurements, the secondary dimensions of the relative wall thickness (RWT%) [22], left ventricular mass (LVM), and left ventricular muscle mass index (LVMI) were calculated [23]. The RWT was obtained as follows: RWT = 2 × PWT/LVEDd. The LVMI was calculated relative to the body surface area (BSA) and was expressed as LVM/BSA. To assess cardiac hypertrophy, LVMI values above 134 g/m^2^ were adopted, as recommended by Levy et al. [24]. Left ventricular (LV) systolic function was determined by measurements of left ventricular ejection fraction (LVEF), and LV diastolic function was evaluated by Doppler echocardiography based on the ratio between the maximal flow velocities during the early diastolic rapid filling phase (mitral inflow *E)* and the late diastolic atrial contraction (*A*-wave velocities) (*E/A*), and the *E/A* ratio was calculated [25]. The intra-observer coefficients of variation were 2.5% for measurements of LV diameters and 3.6% for measurements of LVEF. The echocardiographic measurements were performed by the same operator.

#### 2.2.2. Speckle Tracking Echocardiography

To assess left ventricular strain and rotation, STE was performed using the Vivid E9 device. To analyze cardiac rotation, the following were calculated: the basal rotation (BR) index, the BCS index, and the global longitudinal strain (GLS) of the left ventricle. Two-dimensional images were recorded from the two-, three- (alternatively LAX—left parasternal long axis view), and four-chamber apical views and in the short axis view at the base level of the left ventricle. Digital images to perform the values of strain and rotation were automatically provided by the workstation for analysis. After outlining the left ventricular endocardium in all recorded projections, the computer marked the left ventricle muscle area and divided it into 17 segments and then plotted the curves of left ventricular strain and rotation. GLS was determined from the two-, three- (alternatively LAX – left parasternal long axis view), and four-chamber apical projections and BCS and BR from images in the short axis (Figure 1a–c). The calculated mean systolic peak values of the strain and rotation from all the imaged segments were used for analysis. BR and longitudinal and circumferential strains had all negative values. The strain curves generated from segments of insufficient imaging quality were excluding from analyses (10% of subjects).

##### Observer Variability

The intra-observer variabilities of the measurements of myocardial strain and rotation parameters were examined in blinded fashion in 10 of randomly selected subjects (42 images total). The same operator performed the measurements, and the variability was assessed as the mean percent.

### 2.3. Exercise Protocol 

The subjects underwent an incremental stress test on a cycling ergometer (Sport Excalibur, the Lode Company, Groningen, The Netherlands) to determine individual training indices: maximum power (Pmax) and maximum oxygen uptake (VO_2_max). The subjects started pedaling without any load for 3 min at a speed of 60–70 rpm, and then the load was increased every 3 min by 40 W until the maximum individual load was reached. Heart rate (HR), oxygen uptake (VO_2_), carbon dioxide output (VCO_2_), and respiratory minute volume (V_E_) were continuously monitored before and during the exercise test. Breathing parameters were measured with the Oxycon ALPHA (Jaeger, Höchberg, Germany). HR (PE-3000 Sport-tester, Polar Inc., Kempele, Finland) and systolic and diastolic blood pressures (SDB/DBP) were measured (HEM-907 XL, Omron Corporation, Tokyo, Japan) before and immediately after the test.

### 2.4. Statistical Analysis 

One-way analysis of variance was used to compare anthropometric and physiological variables between the studied groups. The remaining data was analyzed by means of repeated measures analysis of variance (ANOVA). Moreover, research shows that about 28% of athletes presented with left ventricular hypertrophy according to echocardiogram parameters (left ventricular mass index > 134 g/m^2^, relative wall thickness > 0.42 mm) [26]. For male cyclists we expected LVMI increase in 48% of athletes; therefore, a sample size of 42 participants is needed to reach 80% power with two-sided test and alpha of 0.05. The total sample size was also calculated using Altman nomogram and alfa value of 0.05 for 0.07 test power. Homogeneity of variance was verified by Levene’s test. In the absence of homogeneity, Kruskal–Wallis test was used. The significance of differences between the variables was verified with a post-hoc Bonferroni test. Correlation coefficients between all the variables were determined with Pearson’s rank order test. Microsoft Excel 2007 (Microsoft Corp., Washington, USA) and The Statistics Package v.12 (StatSoft Poland, 12.0, Kraków, Poland) were used for data processing and analyses. The differences were considered statistically significant at *p* < 0.05.

## 3. Results

### 3.1. Participants’ Characteristics

Neither BMI, BSA, or any somatic variables were significantly different at baseline (*p* > 0.05) (Table 1). There were significant differences in fat mass content between the athletes and the CG (*p* < 0.05). The LVH and non-LVH athletes were characterized by significantly higher maximum oxygen uptake (VO_2_max) in comparison with the CG (*p* < 0.01 and *p* < 0.05, respectively). Significant differences were found in Pmax between athletes with LVH and non-LVH athletes (*p* < 0.05) and between the LVH and CGs (*p* < 0.01) (Table 1).

### 3.2. Echocardiography

Per the inclusion criteria, athletes with LVH had a significantly higher LVM and LVMI in comparison with the non-LVH and CG groups (*p* < 0.001). ANOVA showed that training significantly increased IVS thickness (F = 14.54; *p* < 0.001) and PWT (F = 6.51; *p* < 0.05) in LVH group. Higher values of PWT and IVS were reported in the LVH group relative to the non-LVH group and the CG (Table 2). The analysis of variance showed a significant effect of LVH (F = 8.96; *p* < 0.001) on LVEDd, which was larger in the LVH group than in the non-LVH group (*p* < 0.05) as well as the CG (*p* < 0.05) (Table 2). Significantly higher RWT and SVs were found in the LVH group relative to the CG (*p* < 0.05). No significant differences were found in LVESd, LVEF, resting HR, SBP/DBP, mitral inflow *E-* and *A*-wave velocities, or the *E/A* ratio between LVH and non-LVH groups, or between non-LVH and CG (Table 2).

### 3.3. STE

The LVH group demonstrated a significantly lower BR at rest in comparison with the non-LVH group (*p* < 0.05) and the CG (*p* < 0.05). Physical effort had no significant effect on BR, nor was this effect different between groups (*p* > 0.05) (Figure 2). ANOVA showed a significant combined influence of LVH and physical effort on BR (F = 5.70; *p* < 0.05) and BCS (F = 4.97; *p* < 0.05). There were no significant differences in BCS at rest between the groups (Figure 3). The two athlete groups showed lower post-effort differences in BCS (0.67% and 0.46% vs. 6.44%) than the CG (Table 3). A significantly higher BCS after exercise in the LVH group was demonstrated in comparison with the CG (*p* < 0.05) (Figure 3). Analysis of variance showed a statistically significant interaction of LVH and physical exercise on GLS (F = 4.59; *p* < 0.05). Statistically significant differences in GLS were observed after exercise between the LVH group and both the non-LVH group (*p* < 0.05) and CG (*p* < 0.05) (Figure 4).

There was a positive correlation between BR at rest and LVM (*r* = 0.64; *p* = 0.035) and a positive correlation between BR and LVMI (*r* = 0.64; *p* = 0.033) (Table 4). Additionally, a trend of a positive correlation between post-exercise BR and LVM (*r* = 0.65; *p* = 0.06) and between BR and LVMI (*r* = 0.66; *p* = 0.06) was revealed. In the group of athletes, a positive correlation between BR and LVEDd at rest (Table 4) and between BR and LVEDd after effort (*r* = 0.73; *p* = 0.025) was also shown. BCS was correlated with SV (*r* = 0.64; *p* = 0.033) and LVEDd (*r* = 0.64; *p* = 0.038) at rest (Table 4).

The values of intra-observer variability in analysis of subjects’ recordings were 7% to 5% for mean basal rotation strain, 6% to 5% for the mean longitudinal strain, and 8% to 5% for global longitudinal strain.

## 4. Discussion

LVH induced by intensive endurance training significantly influenced the basal strain and rotation STE parameters of the heart in competitive athletes; strenuous physical effort also induced lower BR and GLS values in athletes with LVH. The LVH group had lower BR than the other two groups at rest and after physical effort. It is interesting that physical effort did not increase the BR range in our population, though a significant influence of LVH and physical effort on BCS was demonstrated. The LVH group showed lower differences in BCS after effort compared with resting BCS than did the control group. Similarly, smaller GLS values and no differences between GLS at rest and after effort were found in athletes with LVH. The resting BR positively correlated with both LVM and the LVMI; also, specifically for the LVH group, LVEDd positively correlated with BR at rest and after effort.

This study included an assessment of global systolic function of the left ventricle using STE, while taking into account structural differences representing adaptation to intense physical effort [27]. “Strain” and “strain rate” are among the most commonly studied parameters of heart muscle deformation [19,28]. Strain represents the percentage change of the tissue length with reference to its initial length, such that positive values indicate stretching and negative values indicate contraction. The strain rate is the rate of regional deformation expressed as the difference in change velocities between two points. A myocardial contraction results not only from the simple medial movement of the ventricular walls but also from the three-dimensional rotational movement of the ventricle. When viewed from the heart apex, BR occurs in a clockwise direction (i.e., negative values), whereas apical rotation is counterclockwise and generates positive values. The LV rotation corresponds to the absolute difference in the rotation value at both levels and is expressed in degrees [29].

Relatively few studies have been devoted to evaluating the complex biomechanics of athletes’ hearts. However, a few studies comparing athletes with less-active individuals suggested that by inducing cardiomyocyte hypertrophy and affecting strain forces during muscle contractions, physical training caused changes in the components of the longitudinal and circumferential strain of the heart [30]. One study noted a significant effect of training on reducing the range of BCS, but no differences in GLS [31]. It should be emphasized that this research did not account for physiological heart hypertrophy, although there was a suggested relationship between wall thickness and LV radius [32].

The results described above are supported by the data herein, which indicate that the differences between athletes and relatively sedentary individuals are in BR but not longitudinal strain. The analysis of basal strain clearly indicates smaller ranges of values in athletes with LVH both at rest and (more importantly) in response to intense physical effort. Indeed, intense physical training lowered three of the analyzed basal components of left ventricular strain (i.e., longitudinal, circumferential, and radial). Adaptation to physical effort in the longitudinal strain of the heart is no different in athletes, but those in RSs are lower in athletes [31].

A previous study confirming these results has suggested that intense training leading to left ventricular hypertrophy does not change the degree of basal rotation or basal radial strain, while the athletes have a significant increase in apical rotation [33]. However, we can find in the literature reverse data pointing to a greater degree of basal rotation of elite soccer player vs. healthy control (mean: −3.91 vs. −2.61 *p* < 0.03) [34]. Other reports indicate that in athletes with significantly greater LVM compared to untrained individuals (mean: 163 g vs. 111 g), there is no difference in twisting degree between groups [35]. It seems that larger adaptive changes in left ventricular biomechanics might be related to the diastole–untwisting rate [21].

The results of presented study regarding the basal rotation or strain are difficult to explain. It could be speculated that in athletes, the differences in BR observed between LVH and non-LVH groups are only induced by myocardial hypertrophy. The results regarding the effect of cardiac hypertrophy on the basal rotation changes presented in the literature data are different. It has been suggested that in patients with hypertrophic cardiomyopathy, the degree of basal rotation is at least as high as in healthy people without hypertrophy, and even often significantly larger [36,37]. It can therefore be assumed that basal rotation in hypertrophic cardiomyopathy in response to exercise stress can be reduced. It should be pointed out that the distinction of hypertrophy resulting from adaptation of the heart to physical training in relation to primary hypertrophic cardiomyopathy is not easy. The above suggestions may indicate the benefits of evaluating basic left ventricular rotation during physical exercise, in particular the occurrence of mechanical contraction in hypertrophic cardiomyopathy [38].

There is limited explanation regarding the mechanism that determines differences in athletes’ myocardial function. One suggestion is differential stimulation of the autonomic system in athletes versus non-athletes [27]. For example, the lower HR in people with physiological LVH, a result of vasovagal stimulation and differences in sympathetic and parasympathetic stimulation in physical effort, can explain the smaller strains in athletes. However, it remains unclear how a contraction of the variously arranged myofibrils of the heart muscle (subendocardial and subepicardial) influences cardiac shear strain formation. There are few reports on the influence of exercise on the different contractility of the various myocardial fiber layers. Studies on isolated cardiomyocytes from laboratory animals demonstrated a greater effect of physical training on the contractility of the subendocardial layer than on the subepicardial layer [11,39]. However, it is possible that the analyzed indicators were also influenced by the secondary load on the LV and increased systemic vascular resistance. Indeed, it appeared that the lower DBP and absence of differences in the HRpeak value precluded the systemic effect. Other hypotheses include the concept that reduced adrenergic stimulation at rest induces a smaller range of rotation in adaptive cardiac hypertrophy and that reduced shear strain at rest allows the use of a reserve of cardiac rotation during physical effort [31]. A greater reserve could be used to improve systolic function, i.e., the mechanism needed for filling the chambers in the diastolic phase, and thus it could be a key mechanism underlying the increased heart function needed during physical effort [40].

This study found no significant differences in the basal strain of the heart and the global index of the heart rotation during effort relative to rest, which is contradictory to the conceptual framework presented above. Understanding the mechanism responsible for cardiac strain in persons with LVH during effort requires further detailed research; however, one suggestion is that the thickening of the left ventricular walls, especially the interventricular septum, causes “stiffening” of myofibrils and a smaller range of the strain. Knowledge of the mechanisms regulating processes of cardiac remodeling indicated an important role of biological factors in the cardiovascular system [41,42]. An increase in serum somatomedins levels was demonstrated in response to increased activity of the adrenergic system and the renin–angiotensin–aldosterone system, but in pathological heart hypertrophy [43]. It is notable that, despite similar loads and experience in the studied athletes, not all demonstrated LVH. The changes found in the present study clearly suggest adaptive, i.e., physiological, heart hypertrophy [44,45]. Previous studies indicated that serum anabolic hormones elevation, apparently caused by muscle work, might play a crucial role in stimulating myocardial adaptive mechanisms in athletes [45,46,47,48]. However, more studies are needed to investigate whether differences in the hormones responses to effort in athletes stimulate physiological LVH and smaller range of the strain.

## 5. Conclusions

In conclusion, we found that LVH in endurance training had a significant effect on reducing the range of basal rotation of the left ventricle at rest and during physical effort. Left ventricular BR as well as longitudinal and circumferential strains showed less of a difference between rest and after physical effort in subjects with LVH. The obtained results may suggest a reduced adaptive myocardial capacity in terms of the strain and rotation of the heart in athletes with significant myocardial hypertrophy. Echocardiographic assessment of left ventricular rotation using STE can be important in predicting cardiac disorders caused by physical effort in individuals with physiological and pathological heart hypertrophy.

## 6. Limitations of the Study 

There are several limitations but also some favorable aspects to our study. The main limitation of this study is that the lack of the systolic and diastolic LV strain rate parameters. Moreover, untwisting rate should also been used to described the rate of untwisting during the early diastole. However, the left ventricle systolic function determined by investigating the basal component of strain and protocol characteristics allowed an objective analyses and important conclusions. The sample size was limited, which was due to excluding from analysis the strain curves generated from segments of insufficient imaging quality. It was assumed in this study that in the recorded echocardiographic projections for each subject, the number of segments with insufficient imaging quality can be up to two segments per projection and up to four for the whole projection. Analyzing the echocardiogram images recorded in this study, it was found that the insufficient imaging quality concerned only the apical projections. Since systolic strain parameters can be influenced by different factors, we tried to adopt appropriate exclusion criteria. Approximately half of control subjects were recruited from students of Academy of Physical Education and may therefore exhibit better cardiovascular performance than the normal sedentary population. This may possibly have reduced differences between control group and athletes without LVH, although both groups were well-matched regarding age and somatic variables.

## Figures and Tables

**Figure 1 jcm-08-00687-f001:**
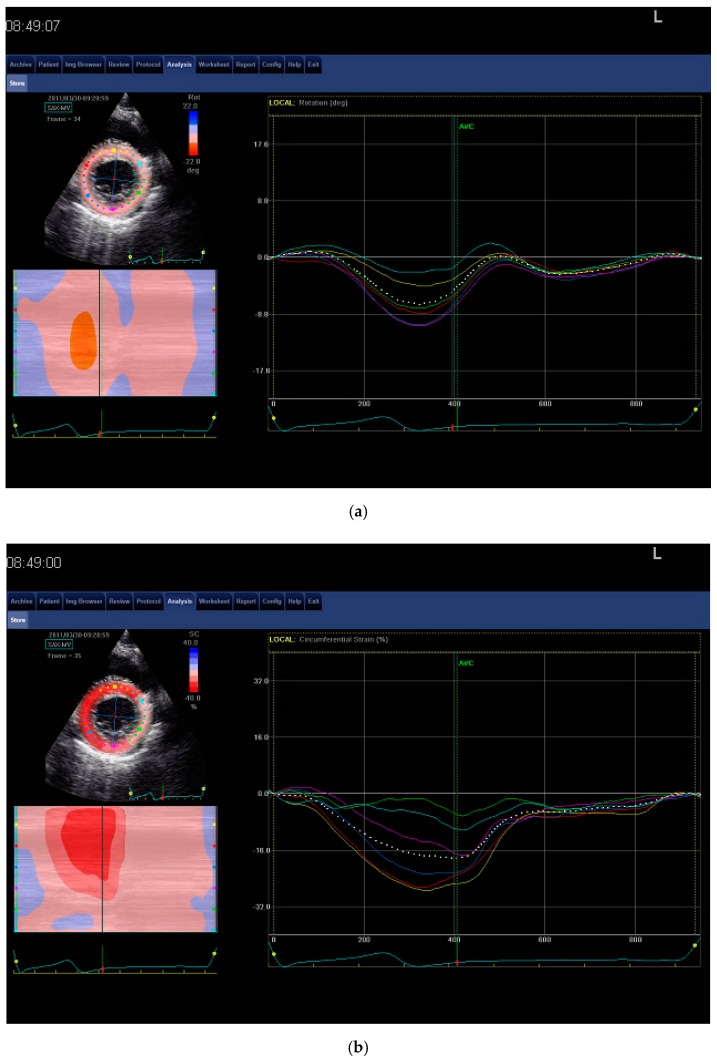
Images of speckle tracking echocardiography. (**a**) Basal rotation of left ventricle (BR)—short parasternal axis at the mitral valve/base level is shown. (**b**) Basal circumferential strain of left ventricle (BCS)—short parasternal axis at the mitral valve/base level is shown. (**c**) Global longitudinal strain of left ventricle (GLS): (C1) longitudinal strain in 4Ch; (C2) longitudinal strain in 2Ch view; (C3) longitudinal strain in left parasternal long axis view (LAX) view; and (C4) bull’s eye mapping analysis.

**Figure 2 jcm-08-00687-f002:**
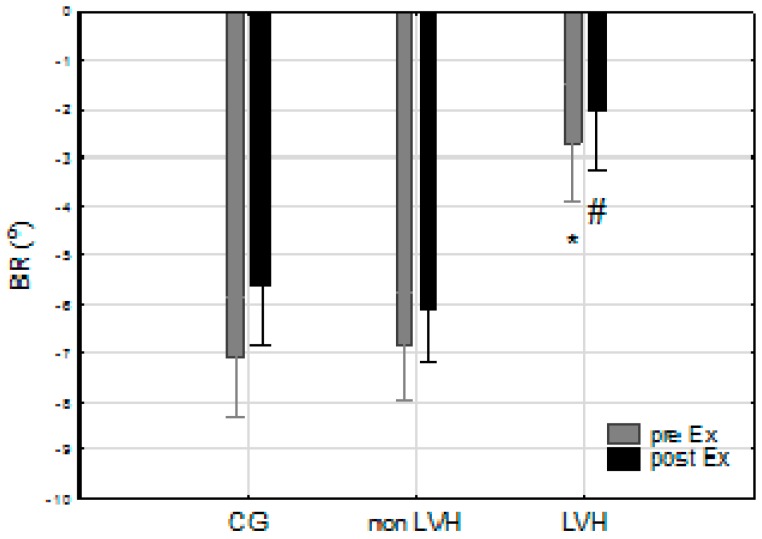
Basal rotation (BR) at rest (Rest) and after maximal exercise (Post-Ex) in the CG, among cyclists without ventricular hypertrophy (non-LVH) and with LVH. * *p* < 0.05 LVH max vs CG max; # *p* < 0.05 LVH vs non-LVH.

**Figure 3 jcm-08-00687-f003:**
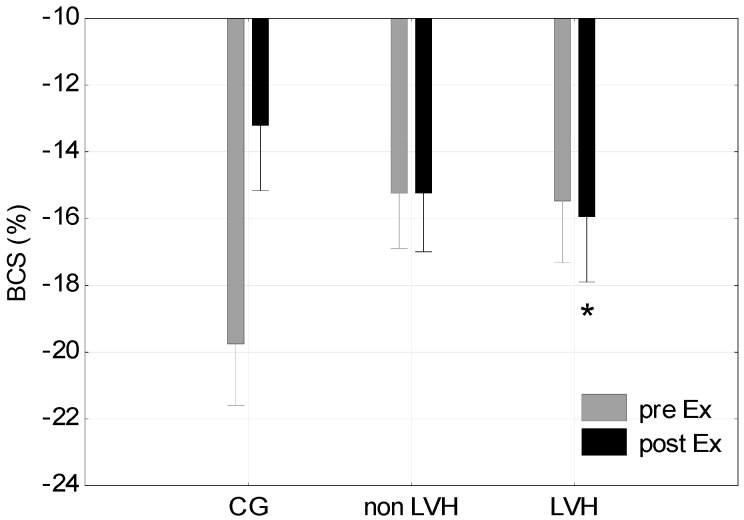
Basal circumferential strain (BCS) at Rest and Post-Ex in the CG, among cyclists without non-LVH and with LVH. * *p* < 0.05 LVH max vs CG max.

**Figure 4 jcm-08-00687-f004:**
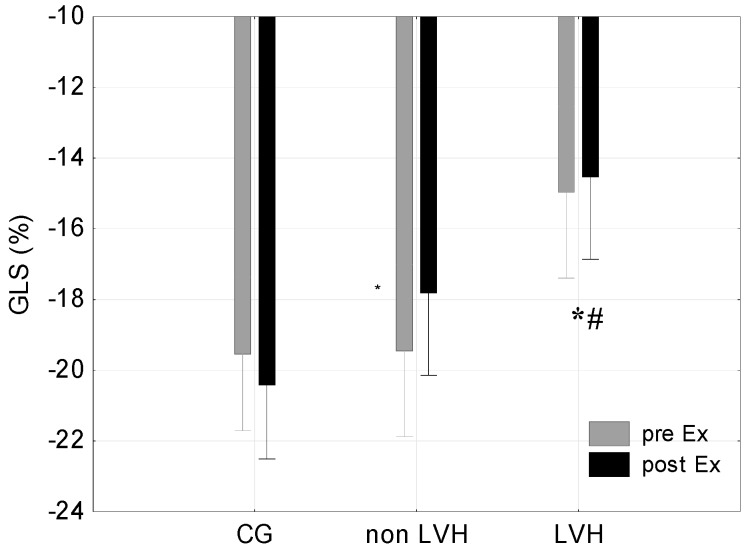
Global longitudinal strain (GLS) at Rest and Post-Ex in the CG, among cyclists without ventricular hypertrophy (non-LVH) and with LVH. * *p* < 0.05 LVH vs CG; # *p* < 0.05 LVH vs non-LVH.

**Table 1 jcm-08-00687-t001:** Characteristics of study participants: athletes with left ventricle hypertrophy (LVH), athletes without left ventricle hypertrophy (non-LVH), and untrained control group (CG).

Variable	CG (*n* = 14)	Non-LVH (*n* = 14)	LVH (*n* = 14)
Age (years)	20.13 (0.4)	27.21 (12.4)	24.43 (3.5)
Body height (cm)	178.38 (4.9)	177.43 (3.6)	179.39 (7.3)
Body mass (kg)	75.87 (7.7)	70.70 (5.0)	71.96 (7.3)
BSA (m^2^)	1.94 (0.1)	1.88 (0.1)	1.90 (0.1)
BMI (kg/m^2^)	23.85 (2.4)	22.38 (1.7)	22.37 (2.0)
FM (kg)	11.72 (2.9)	8.50 (3.2) *	8.58 (2.7) **
FFM (kg)	62.77 (4.6)	61.46 (4.6)	65.96 (6.3)
TBW (kg)	45.85 (3.4)	46.78 (4.3)	48.35 (4.6)
VO_2_max (mL/kg/min)	50.33 (4.8)	62.18 (7.8) *	62.00 (7.5) **
Training status (years)	0	9.10 (4.3)	9.90 (5.1)
Power_max_ (Watt)	260 (40.0)	368.89 (36.67) *	400 (34,11) **#

BSA—body surface area; BMI—body mass index; FM—fat mass; FFM—fat free mass; TBW—total body water; VO_2_max—maximal oxygen uptake. * *p* < 0.01 non-LVH vs CG; ** *p* < 0.01 LVH vs CG; and # *p* < 0.05 LVH vs non-LVH.

**Table 2 jcm-08-00687-t002:** Echocardiographic variables and blood pressure in athletes with left ventricle hypertrophy (LVH), athletes without left ventricle hypertrophy (non-LVH), and untrained control group (CG).

Variable	CG (*n* = 14)	Non-LVH (*n* = 14)	LVH (*n* = 14)
LVM (g)	159.04 (21.2)	196.20 (33.8)	301.29 (58.4) ***###
LVMI (g/m^2^)	82.07 (10.5)	104.54 (17.2)	158.04 (25.6) ***###
IVS (mm)	8.69 (1.0)	10.18 (2.0)	12.14 (0.7) ***##
PWT (mm)	8.5 (1.2)	9.18 (1.6)	11 (1.1) ***#
RWT (%)	36.00 (6.0)	38.00 (10)	41.00 (5.0) *
LVEDd (mm)	47.88 (2.1)	49.07 (3.9)	53.93 (4.2) *#
LVESd (mm)	28.63 (2.1)	30.71 (2.9)	31.00 (3.5)
LVEF%	64.50 (5.3)	60.36 (5.4)	60.57 (4.2)
SV (mL)	77.00 (7.1)	86.00 (12.2)	94.36 (10.1) *
HR_Rest_ (b/min)	73.00 (7.5)	63.00 (2.5)	60.00 (3.0)
HR _Peak_ (b/min)	184.00 (7.1)	185.00 (8.0)	192.00 (3.0)
SBP_Rest_ (mmHg)	126.86 (10.7)	118.57 (13.9)	124.64 (14.1)
SBP _Post-Ex_ (mmHg)	182.90 (8.1)	185.7 (13.6)	196.2 (7.4)
DBP_Rest_ (mmHg)	81.25 (11.3)	78.57 (12.2)	77.50 (11.2)
DBP _Post-Ex_ (mmHg)	64.50 (10.0)	60.00 (6.7)	57.10 (11.1)
E/A	1.98 (0.6)	1.80 (0.5)	1.91 (0.3)
E (m/s)	0.91 (0.1)	0.90 (0.1)	0.90 (0.2)
A (m/s)	0.48 (0.1)	0.52 (0.1)	0.47 (0.05)

LVM—left ventricular mass; PWT—posterior wall thickness RWT%—relative wall thickness; LVEDd—left ventricular end-diastolic diameter; LVESd—left ventricular end-systolic diameter; LVEF—left ventricular ejection fraction; SV—stroke volume; HR_Rest_—resting heart rate and at maximal exercise intensity (Peak); SBP_Rest_—systolic blood pressure at rest and after maximal exercise (Post-Ex); DBP_Rest_—diastolic blood pressure at rest and after maximal exercise (Post-Ex); E/A—mitral inflow assessment. * *p* < 0.05; *** *p* < 0.001 LVH vs CG; # *p* < 0.05; # # *p* < 0.01; and # # # *p* < 0.001 LVH vs non-LVH.

**Table 3 jcm-08-00687-t003:** Basal rotation absolute value (BR) and relative changes of basal circumferential strain (BCS) and GLS at Rest and Post-Ex and the dynamics of changes (Δ) in the athletes with LVH, athletes without left ventricle hypertrophy (non-LVH) and untrained CG.

Indicator	CG			Non-LVH			LVH		
	Rest	Post-Ex	Δ	Rest	Post-Ex	Δ	Rest	Post-Ex	Δ
Basal rotation (BR) (°)	−7.09 (2.86)	−5.63 (2.30)	1.47 (3.96)	−6.84 (2.12)	−6.09 (3.18)	0.76 (1.69)	−2.11 (0.60) *#	−2.02 (0.51) *#	0.69 (2.36)
Basal circumferential strain (BCS) (%)	−19.75 (4.28)	−13.21 (2.82)	6.55 (3.59)	−15.91 (1.96)	−15.24 (5.28)	0.67 (4.47)	−15.47 (3.52)	−15.93 (1.94) *	0.46 (4.60)
Global longitudinal strain (GLS) (%)	−19.54 (5.88)	−20.42 (0.96)	−0.88 (1.72)	−19.45 (2.73)	−17.82 (2.75)	1.64 (0.29)	−14.97 (4.75)	−14.54 (2.64) *#	0.43 (1.57)

The changes considered for basal rotation were absolute changes and that the changes considered for longitudinal and circumferential strains were relative changes * *p* < 0.05; LVH vs CG; # *p* < 0.05; and LVH vs non-LVH.

**Table 4 jcm-08-00687-t004:** Correlations between echocardiographic variables and speckle tracking echocardiography (STE) measures at rest.

Variables	BR Rest	BR Post-Ex	BCS Rest	BCS Post-Ex	GLS Rest	GLS Post-Ex
LVM (g)	*r* = 0.64 *p* < 0.05	*r* = 0.65 *p* > 0.05	*r* = 0.36 *p* > 0.05	*r* = −0.01 *p* > 0.05	*r* = 0.08 *p* > 0.05	*r* = 0.30 *p* > 0.05
LVMI (g/m^2^)	*r* = 0.64 *p* < 0.05	*r* = 0.66 *p* > 0.05	*r* = 0.48 *p* > 0.05	*r* = 0.01 *p* > 0.05	*r* = 0.02 *p* > 0.05	*r* = 0.17 *p* > 0.05
LVEDd (mm)	*r* = 0.61 *p* < 0.05	*r* = 0.73 *p* < 0.05	*r* = 0.64 *p* < 0.05	*r* = 0.39 *p* > 0.05	*r* = 0.11 *p* > 0.05	*r* = 0.27 *p* > 0.05
LVESd (mm)	*r* = 0.48 *p* > 0.05	*r* = 0.41 *p* < 0.05	*r* = 0.41 *p* > 0.05	*r* = 0.26 *p* > 0.05	*r* = 0.09 *p* > 0.05	*r* = 0.19 *p* > 0.05
SV (mL)	*r* = 0.20 *p* > 0.05	*r* = 0.34 *p* > 0.05	*r* = 0.64 *p* < 0.05	*r* = 0.12 *p* > 0.05	*r* = −0.23 *p* > 0.05	*r* = −0.24 *p* > 0.05
LVEF (%)	*r* = 0.49 *p* > 0.05	*r* = 0.36 *p* > 0.05	*r* = −0.01 *p* > 0.05	*r* = −0.20 *p* > 0.05	*r* = 0.40 *p* > 0.05	*r* = 0.49 *p* > 0.05

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
