# Peer review of "Left Ventricular Systolic Function Assessed by Speckle Tracking Echocardiography in Athletes with and without Left Ventricle Hypertrophy"

_jcm, 2019, doi:10.3390/jcm8050687_

Reviewer 1 Report

Manuscript entitled “Left ventricular function assessed by speckle tracking echocardiography in athletes with and without left ventricle hypertrophy”

 Authors: Aleksandra Zebrowska et al.

 Summary: The revised version of the paper by Zebrowska et al. improved significantly, as the authors fulfilled this reviewer’s requirements and answered the questions properly. Authors provided illustrative figures, updated Methods and Discussion sections. Echocardiographic parameters were added, and relative wall thickness (RWT%) was calculated for each subject, further supporting previous findings.

In summary, as the authors carefully revised and significantly updated the manuscript.

Author Response

Response to Reviewer

We appreciate the time and efforts by the editor in reviewing the manuscript title: “Left ventricular function assessed by speckle tracking echocardiography in athletes with and without left ventricle hypertrophy” send to Journal of Clinical Medicine.

Reviewer 2 Report

The authors have significantly improved their manuscript and have taken into account all the comments of the reviewers. Nevertheless, I still have some other comments.

1. Please further detail how you calculated the power of your study in order to precisely define your sample size.

2. I understood the reason why you have assessed the function of the whole left ventricle only by investigating the basal component of strain. Nevertheless, this is an important limitation of your study, which should be added in the limitations section of the manuscript.

3. You said that “diastolic strain rate and untwisting rate" are rarely evaluated and don’t provide additional valuable information due to the fact that these variables are resulted of mathematical transformation of the initial parameters - eg diastolic strain”. I disagree with the authors. First, untwisting rate is the key parameter for assessing the LV diastolic dysfunction in the vast majority of studies in patients with heart failure with preserved ejection fraction or with hypertrophic cardiomyopathy (Notomi Circulation 2006, Wang Circulation 2007, Notomi Am J Physiol Heart Circ Physiol 2008, Burns JACC 2009…). Moreover, the strain rate parameters, even if they result from mathematical transformation of the initial strain parameters, provide very interesting data regarding the velocity of the systolic and diastolic myocardial LV deformation. Some authors even suggest that systolic strain rate is a better parameter of contractile LV function than systolic strain. Thus, the fact that the systolic strain rate was not investigated in order to assess the systolic LV function should also be added in the limitations section. 

4. Please specify in the methods section that the values of strain/rotation you considered are the values automatically provided by the workstation (as suggested by the different figures). 

5. As you state that “We would like to suggest, that diastolic function assessment was not a main purpose of our study”, please delete the assessment of the LV diastolic function, since the E/A ratio is only a poor surrogate of the LV diastolic function, especially in, patients with preserved ejection fraction (Nagueh and colleagues, European Heart Journal – Cardiovascular Imaging (2016) 17, 1321–1360). In this regard, please change the title to clearly indicate to the readers that you focus on the LV systolic function only. If you also want to assess the LV diastolic function, please use adequate parameter (e’ wave and E/e’ ratio) and provide the diastolic strain rate parameters. Also change the following sentence in the discussion page 10, line 6: “This study included an assessment of global systolic and diastolic function of the left ventricle” since the diastolic function is not adequately assessed. In addition, as I asked you in my first reviewing, the sentence must also be changed because you only investigated the basal component of rotation and circumferential strain.

6. Please add in the methods section the formula used to calculate the intra-observer variability of measurements. Please also add that the echocardiographic measurements were performed by the same operator.

7. You indicate in Table 2 that the heart rate after exercise was around 190 bpm in the different groups of subjects. Are you really sure of these very high values? It is unexpected, especially in athletes.

8. Please add in the legend of Table 3 when you consider absolute or relative changes.

9. The references 17 and 22 must be updated.

Author Response

Response to Reviewer: 2

We appreciate the time and efforts by the editor in reviewing the manuscript title: “Left ventricular function assessed by speckle tracking echocardiography in athletes with and without left ventricle hypertrophy” send to Journal of Clinical Medicine. We have addressed all issues indicated in the review report, and believed that the revised version can meet the journal publication requirements. Your suggestions have been of high value and we have taken them into account. The following corrections have been attempted (all of them have been marked red).

Comments and Suggestions for Authors

The authors have significantly improved their manuscript and have taken into account all the comments of the reviewers. Nevertheless, I still have some other comments.

1. Please further detail how you calculated the power of your study in order to precisely define your sample size.

Answer: We appreciate this comment. The total sample size was calculated using Altman nomogram and alfa value of 0.05 for 0.07 test power. We would apologize for this mistake and we have changed the sentences “Power calculation” to “The total sample size”.

The selected publications on sample size calculation: Whitley, E.; J. Statistics review 4: Sample size calculations. Critical Care. 2002, 6(4), 335–341 and Altman, D.G. How large a sample? In: Gore SM, Altman DG, editor. Statistics in Practice. London, UK: British Medical Association; 1982.

Moreover, research shows that about 28% of athletes presented with left ventricular hypertrophy according to echocardiogram parameters (left ventricular mass index>134g/m2, relative wall thickness >0.42 mm) (Samesina et al. 2017). For male cyclists we expected LVMI increase in 48 % of athletes, therefore a sample size of 42 participants is needed to reach 80% power with two-sided test and alpha of 0.05.

Samesina N.; Azevedo, L.F,; DE Matos, LDNJ,; Echenique, LS,; Negrao, CE,; Pastore, CA. Comparison of Electrocardiographic Criteria for Identifying Left Ventricular Hypertrophy in Athletes from Different Sports. Clinics 2017, 72(6), 343-350.

 2. I understood the reason why you have assessed the function of the whole left ventricle only by investigating the basal component of strain. Nevertheless, this is an important limitation of your study, which should be added in the limitations section of the manuscript.

Answer: We agree with the expert reviewer and we have added this to the discussion section and we have also written about them in the limitations of the study. In our further research into the LV speckle tracking derived cardiac strain response to physical effort in athletes, we will apply the method rightly recommended by the reviewer.

3. You said that “diastolic strain rate and untwisting rate" are rarely evaluated and don’t provide additional valuable information due to the fact that these variables are resulted of mathematical transformation of the initial parameters - eg diastolic strain”. I disagree with the authors. First, untwisting rate is the key parameter for assessing the LV diastolic dysfunction in the vast majority of studies in patients with heart failure with preserved ejection fraction or with hypertrophic cardiomyopathy (Notomi Circulation 2006, Wang Circulation 2007, Notomi Am J Physiol Heart Circ Physiol 2008, Burns JACC 2009…). Moreover, the strain rate parameters, even if they result from mathematical transformation of the initial strain parameters, provide very interesting data regarding the velocity of the systolic and diastolic myocardial LV deformation. Some authors even suggest that systolic strain rate is a better parameter of contractile LV function than systolic strain. Thus, the fact that the systolic strain rate was not investigated in order to assess the systolic LV function should also be added in the limitations section.  

Answer: Thank you. We have changed the text to be more precise in our statements. We totally agree that measures of LV twist mechanics (basal rotation but also apical rotation, twist and untwisting rate/velocity) provide very interesting data regarding LV deformation. However, in a meta-analysis performed by Beaumont et al. 2017, it should be noted that there is not one established preferred STE protocol to analyse the LV physiological remodelling as a consequence of repetitive cardiac loading. We agree with your suggestions that these parameters may add important value to our study and we will investigate these values in our further observations.

The fact that the systolic strain rate was not investigated in order to assess the systolic LV function in our study was added in the limitations section.  

4. Please specify in the methods section that the values of strain/rotation you considered are the values automatically provided by the workstation (as suggested by the different figures).  

 Answer: We agree with the expert reviewer and we have added all of these details to the methods section.

5. As you state that “We would like to suggest, that diastolic function assessment was not a main purpose of our study”, please delete the assessment of the LV diastolic function, since the E/A ratio is only a poor surrogate of the LV diastolic function, especially in, patients with preserved ejection fraction (Nagueh and colleagues, European Heart Journal – Cardiovascular Imaging (2016) 17, 1321–1360). In this regard, please change the title to clearly indicate to the readers that you focus on the LV systolic function only. If you also want to assess the LV diastolic function, please use adequate parameter (e’ wave and E/e’ ratio) and provide the diastolic strain rate parameters. Also change the following sentence in the discussion page 10, line 6: “This study included an assessment of global systolic and diastolic function of the left ventricle” since the diastolic function is not adequately assessed. In addition, as I asked you in my first reviewing, the sentence must also be changed because you only investigated the basal component of rotation and circumferential strain.

Answer: We agree with the expert reviewer and the following corrections have been made:

We delated the “assessment of the LV diastolic function (page, line)

We changed the title “Left ventricular systolic function assessed by speckle tracking echocardiography in athletes with and without left ventricle hypertrophy”.

We changed the sentences in the Discussion section “This study included an assessment of global systolic function of the left ventricle”

6. Please add in the methods section the formula used to calculate the intra-observer variability of measurements. Please also add that the echocardiographic measurements were performed by the same operator.

 Answer: We agree with the expert reviewer. We have added all of these details to the Methods section.

7. You indicate in Table 2 that the heart rate after exercise was around 190 bpm in the different groups of subjects. Are you really sure of these very high values? It is unexpected, especially in athletes.

Answer: We greatly appreciate the Reviewer’s comments. Heart rate was assessed at maximal exercise intensity (HRpeak value) but not immediately post-exercise.  We have changed this in Table 2 and in the text.

8. Please add in the legend of Table 3 when you consider absolute or relative changes.

Answer: We agree with the expert reviewer and we added this information in the legend of Table 3.

9. The references 17 and 22 must be updated.Oxford Academic PubMed

Google ScholarAnswer: We agree with the expert reviewer and we have updated these publications.

Badano, L.P.; Kolias, T.J.; Muraru, D.; Abraham, T.P.; Aurigemma, G., Edvardsen, T.; D’Hooge, J.; Donal, E.; Fraser, A.G.; Marwick, T., et al. Standardization of left atrial, right ventricular, and right atrial deformation imaging using two-dimensional speckle tracking echocardiography: a consensus document of the EACVI/ASE/Industry Task Force to standardize deformation imaging. European Heart Journal - Cardiovascular Imaging. 2018,19(6), 591–600.

Lang, R.; Badano, L.P.; Mor-Avi, V.; Afilalo, J.; Armstrong, A.; Ernande, L.; Flachskampf, F.A.; Foster, E.; Goldstein, S.A.; Kuznetsova, T., et al.

Recommendations for Cardiac Chamber Quantification by Echocardiography in Adults: An Update from the American Society of Echocardiography and the European Association of Cardiovascular Imaging European Heart Journal - Cardiovascular Imaging. 2015, 16(3), 233–271.

Roberto M. Lang, MD, FASE, FESCChicago, Illinois; Padua, Italy; Montreal, Quebec and Toronto, Ontario, Canada; Baltimore, Maryland; Créteil, France; Uppsala, Sweden; San Francisco, California; Washington, District of Columbia; Leuven, Liège, and Ghent, Belgium; Boston, MassachusettsSearch for other works by this author on:

Oxford Academic

PubMed

Google Schola

Oxford Academic

PubMed

Google Scholar

Chicago, Illinois; Padua, Italy; Montreal, Quebec and Toronto, Ontario, Canada; Baltimore, Maryland; Créteil, France; Uppsala, Sweden; San Francisco, California; Washington, District of Columbia; Leuven, Liège, and Ghent, Belgium; Boston, Massachusetts

Search for other works by this author on:

Oxford Academic

PubMed

Google ScholaRound  2

Reviewer 2 Report

Dear authors, please consider the following three last minor comments.

1. Please add in the methods section, as asked in my previous review, the formula used to calculate the intra-observer variability of measurements for conventionnal and speckle tracking measurements.

2. Please provide the intra-observer coefficient of variation for the different speckle tracking measurements.

3. The legend of Table 3 is still unclear. Please clearly state that the changes you considered for basal rotation were absolute changes and that the changes you considered for longitudinal and circumferential strains were relative changes. By the way, why did you consider absolute or relative changes depending on the different speckle tracking parameters? 

Author Response

Comments and Suggestions for Authors

Dear authors, please consider the following three last minor comments.

1. Please add in the methods section, as asked in my previous review, the formula used to calculate the intra-observer variability of measurements for conventional and speckle tracking measurements.

Answer: We agree with the expert reviewer and the formula used to calculate the intra-observer variability was added.

The intra-observer variabilities of the measurements of myocardial strain and rotation parameters were examined in blinded fashion in 10 of randomly selected subjects (42 images total). The same operator performed the measurements and the variability was assed as the mean percent.

2. Please provide the intra-observer coefficient of variation for the different speckle tracking measurements.

Answer: We agree with the expert reviewer and present the values of intra-observer coefficient of variation for the different speckle tracking measurements.

The values of intra-observer variability in analysis of subjects’ recordings were 7% to 5% for mean basal rotation strain, 6 % to 5 % for the mean longitudinal strain and 8% to 5% for global longitudinal strain.

3. The legend of Table 3 is still unclear. Please clearly state that the changes you considered for basal rotation were absolute changes and that the changes you considered for longitudinal and circumferential strains were relative changes. By the way, why did you consider absolute or relative changes depending on the different speckle tracking parameters? 

Answer: We agree with the expert reviewer and we added the sentence that the changes you considered for basal rotation were absolute changes and that the changes you considered for longitudinal and circumferential strains were relative changes.

The values obtained by STE were presented as absolute or relative. This methods and variation is mainly due to our previous study (Int J Cardiovasc Imaging (2013) 29:797-808 Staron A. et al., Arch Med. Sci (2014) 10: 1091-1100 Liszka et al.) and the different dedicated variables. The strain values are expressed in percentages and rotation is expressed in degrees.